# Genetic Inheritance and the Impact of Low Birth Weight on the Incidence of Cryptorchidism in Hyperprolific Sows

**DOI:** 10.3390/ani15213105

**Published:** 2025-10-25

**Authors:** Thanut Wathirunwong, Padet Tummaruk, Sarthorn Porntrakulpipat, Jatesada Jiwakanon

**Affiliations:** 1Department of Medicine, Faculty of Veterinary Medicine, Khon Kaen University, Khon Kaen 40002, Thailand; thanut.w@kkumail.com (T.W.);; 2Department of Obstetrics, Gynaecology and Reproduction, Faculty of Veterinary Science, Chulalongkorn University, Bangkok 10330, Thailand; 3Centre of Excellence in Swine Reproduction, Chulalongkorn University, Bangkok 10330, Thailand

**Keywords:** birthweight, cryptorchidism, insemination, litter size, pig

## Abstract

**Simple Summary:**

Cryptorchidism, a condition in which one or both testicles fail to descend, is an important problem in pig production because it can reduce meat quality and market value. This study investigated the occurrence of cryptorchidism in piglets and its relationship with piglet birth weight and litter size. Cryptorchid piglets were identified during the first week of life. Results showed that cryptorchid piglets were more frequent in hyperprolific sows and that lower birth weight increased the likelihood of cryptorchidism, whereas litter size itself was not directly associated. These findings emphasize the role of birth weight in the condition and highlight the need for management strategies to reduce low birth weight in hyperprolific sows.

**Abstract:**

Cryptorchidism in piglets, characterized by undescended testicles, causes economic losses and reduces consumer acceptance. Hyperprolific sows (HPS) have been hypothesized to produce a higher incidence of cryptorchid offspring. This study investigated the incidence of cryptorchidism in piglets born to HPS and its association with piglet birth weight and litter size in an observational study. Data from 276 litters (144 Landrace × Yorkshire sows; 4003 piglets) were analyzed. Sows were classified by genetic line (conventional: 68 litters; HPS: 208 litters) and parity (primiparous: 144; second parity: 132). At first parity, all gilts were inseminated with semen from a phenotypically unilateral cryptorchid Duroc boar, whereas at second parity, semen from three normal Duroc boars, which were full siblings, was used. The Landrace × Yorkshire HPS line produced more piglets per litter than the conventional Landrace × Yorkshire line (16.5 ± 0.3 vs. 12.4 ± 0.6; *p* < 0.001). Cryptorchidism occurred in 25.7% (37/144) of litters inseminated with semen from the cryptorchid boar, compared with 3.8% (5/132) of litters inseminated with semen from normal boars (*p* < 0.001). In total, 42 sows produced at least one cryptorchid piglet across both parities. Among affected sows (*n* = 42), the average number of cryptorchid piglets per litter was 1.3 ± 0.6 (range: 1–3). In the HPS line, cryptorchidism was detected in 24.1% (26/108) of litters, compared with 30.6% (11/36) in the conventional line (*p* = 0.441). HPS piglets had lower birth weights than conventional piglets (1.14 ± 0.01 vs. 1.30 ± 0.02 kg; *p* < 0.001). In the HPS line, litters with cryptorchid piglets had lower birth weights than those without (1.11 ± 0.02 vs. 1.18 ± 0.01 kg; *p* = 0.012), whereas no difference was observed in the conventional line (1.31 ± 0.04 vs. 1.28 ± 0.02 kg; *p* = 0.917). Litter size did not differ between litters with and without cryptorchid piglets in either genetic line. In conclusion, the lower average birth weight in cryptorchid litters of the HPS line, but not in conventional lines, suggests that HPS breeds may influence cryptorchidism incidence. These findings highlight the need to optimize fetal growth especially in the HPS to reduce this risk.

## 1. Introduction

Cryptorchidism, a congenital defect in male mammals, occurs when one or both testes fail to descend into the scrotum. In pigs, testicular descent is typically completed around the time of birth, similar to other livestock species [1]. This process involves the growth and expansion of the gubernaculum, beginning primarily with transabdominal testicular migration around day 55 of gestation, influenced by Leydig cell products such as insulin-like peptide 3 [2]. The descent continues with inguinoscrotal migration, which occurs between days 85 and 90 of gestation, facilitated by testosterone [1,3]. Among common livestock species, pigs exhibit one of the highest incidences of cryptorchidism, with prevalence rates reported to range from 2.2% to 12.0% [4,5,6,7]. In affected pigs, the most frequent anatomical location of the retained testis is the abdominal region [7,8].

Failure of testicular migration has long been recognized as a heritable homozygous recessive trait in certain pig breeds [8,9]. It is a complex multigenic condition, with heritability estimates ranging from 0.26 to 0.32, indicating a significant genetic contribution to its occurrence [10]. Practical control strategies include selecting sire lines free of cryptorchidism and excluding affected animals. However, the condition still occurs [4]. Cryptorchidism has significant economic implications, primarily due to the risk of boar taint, an undesirable odor and taste in the meat of affected animals, which can reduce consumer acceptance [11]. To mitigate these economic losses, cryptorchidectomy via laparotomy is commonly performed after weaning, typically around 5–6 weeks of age. However, this surgical procedure incurs additional costs and requires specialized veterinary expertise [7,12]. Alternatively, affected pigs may be sold at a discounted price, further impacting economic losses in the swine industry [13]. Additionally, immunocastration has been proposed as an alternative approach to prevent boar taint and avoid surgical intervention. Nevertheless, in Thailand this strategy is not widely adopted, as pork consumers remain concerned that immunocastrated males may still develop boar taint, making such animals more difficult to market.

Recent studies in mammals have shown that cryptorchidism results from complex interactions among genetic, epigenetic, and environmental factors [14,15]. For example, elevated estradiol levels in the placenta of neonates with cryptorchidism, potentially due to exposure to environmental estrogenic agents, may contribute to the condition through overexpression of the aromatase gene [14]. In humans, several risk factors, including low birth weight, premature birth, low parity, and exposure to environmental pollutants such as pesticides and estrogenic compounds, have been associated with an increased incidence of cryptorchidism [15]. Similarly, in pigs, Dolf et al. [5] reported that larger litter size and higher stillbirth rate were correlated with a higher incidence of cryptorchid piglets. Furthermore, under comparable experimental conditions using a cryptorchid boar, Fredeen and Newman [4] observed variations in the incidence of cryptorchid male piglets among different pig breeds, suggesting breed-specific susceptibility to the condition. To our knowledge, the prevalence of cryptorchidism and its risk factors in modern hyperprolific sow (HPS) genetics, widely adopted in the global swine industry, have not yet been thoroughly investigated.

Modern commercial sows have been genetically selected for high prolificacy traits [16]. HPS is typically defined as one that consistently produces more piglets at birth than the number of its functional teats [17]. In general, sows producing more than 16 piglets per litter are defined as HPS. Due to their large litter sizes, these sows inevitably experience intrauterine crowding during gestation, which often leads to intrauterine growth retardation (IUGR) in fetuses [18,19]. Notably, in pig production, birth weight is commonly used as one of the criteria to identify IUGR, which is typically defined as an absolute birth weight of less than 1.1 kg [20,21]. Recently, abnormal testicular development has been recognized as a common consequence of IUGR, characterized by gonadal dysplasia, reduced testicular volume, and hormonal imbalance [22,23]. These alterations may contribute to impaired testicular descent, thereby potentially increasing the risk of cryptorchidism in male piglets. In addition, HPS farms commonly have litters with lower average birth weights and greater variability in piglet birth weight [24]. To the best of our knowledge, the effects of IUGR and piglet birth weight on the incidence of cryptorchidism in HPS have not been previously reported. Given the established risk factors for cryptorchidism in pigs, it is crucial to assess whether genetic selection for increased prolificacy has influenced its occurrence in commercial swine production. Understanding these potential impacts is essential for developing effective breeding strategies and farm management practices to mitigate cryptorchidism. This study aimed to investigate the incidence of cryptorchidism in HPS and its relationship with piglet birth weight and litter size in a commercial farming setting.

## 2. Materials and Methods

### 2.1. Study Design

This research adhered to the Ethical Principles and Guidelines for the Use of Animals in Scientific Research as specified by the National Research Council of Thailand (NRCT) and was approved by the Institutional Animal Care and Use Committee (IACUC) of Khon Kaen University (protocol number IACUC-KKU-90/67). The study was conducted at a commercial pig farm in Khon Kaen Province, Thailand, from October 2022 to November 2023.

A total of 144 Landrace × Yorkshire crossbred sows were included in the study, consisting of 108 HPS and 36 conventional sows in the first parity. All 108 HPS and 36 conventional gilts were selected from the entire sow population on the farm during the study period based on the reproductive performance records of their parental lines. However, during weaning to the next farrowing, 12 sows were culled due to reproductive failures, including absence of estrus, abortion, complete litter mummification, and sudden death. Therefore, 100 HPS and 32 conventional sows which gave birth in the second parity were used. A total of 276 litters from both parities comprising 4003 live-born piglets, 1988 males and 2014 females, were used in the analysis, one hermaphroditic piglet was identified and excluded from the cryptorchidism calculation due to its ambiguous sex phenotype. Distribution of piglets across genetic lines and parities of sow together with boar types is presented in Table 1.

The sows were classified by genetic line as either conventional Landrace × Yorkshire line (*n* = 68 litters) or the modern Danish Landrace × Yorkshire crossbred HPS line (*n* = 208 litters, DanBred^®^, Vejle, Denmark). The conventional sows line were F1 Landrace × Yorkshire gilts derived from the farm’s long established lines, which historically produced ≤14 total born piglets per litter. In contrast, HPS were F1 Landrace × Yorkshire gilts purchased from DanBred^®^ genetics and were characterized by extremely large litters (>16 piglets), often exceeding teat capacity. The sows were further categorized by parity into primiparous (*n* = 144 litters) and second parity groups (*n* = 132 litters). In parity 1 (P1), all gilts were inseminated with semen from a phenotypically unilateral cryptorchid Duroc boar, whereas in parity 2 (P2), the sows were inseminated with semen from three normal full-sibling Duroc boars.

Data on insemination date, farrowing date, and piglet characteristics were collected. The following reproductive and piglet performance parameters were calculated for statistical analysis: farrowing rate, total number of piglets born per litter (TB), number of piglets born alive per litter (BA), average piglet birth weight (BW), percentage of mummified fetuses per litter (MF), and percentage of stillborn piglets per litter (SB). Cryptorchidism was assessed on a litter basis, with a litter classified as cryptorchid if at least one piglet exhibited cryptorchidism and as normal if no cryptorchid piglets were detected.

The presence of cryptorchidism was evaluated by palpation between days 1 and 7 postpartum. All male piglets were initially examined by trained farm staff under the supervision of a licensed veterinarian within 24 h after birth. Each palpation was immediately rechecked by the veterinarian during the examination to ensure diagnostic accuracy. Piglets in which one or both testes were absent from the scrotum were classified as cryptorchid. To minimize the risk of false negatives, a second examination was performed at weaning (28 days of age), during which cryptorchid piglets underwent surgical removal of the undescended testes if the condition was confirmed. The percentage of cryptorchid piglets was calculated based on the total number of male piglets within each litter and expressed as a percentage.

### 2.2. Semen Preparation and Insemination Techniques

The current study included one 9-month-old Duroc boar with a unilateral cryptorchid phenotype and three 8-month-old phenotypically normal Duroc boars, which were full siblings and had no known history of cryptorchidism in their lineage for at least two generations. Semen was collected from these boars using the gloved-hand method, and semen quality was assessed following standardized on-farm protocols. Only ejaculates meeting the following criteria were used for artificial insemination (AI): semen volume greater than 100 mL, sperm concentration of at least 200 × 10^6^ motile sperm/mL, and subjective sperm motility of 80% or higher. After quality assessment, semen was diluted with Kobidil^®^, a medium-term semen extender (Landata, Saint-Gilles, France). Prepared insemination doses containing 3 × 10^9^ motile sperm in an 80 mL volume were used for conventional intracervical AI.

Heat detection was conducted once daily in the morning by trained personnel employing the back-pressure method alongside a teaser boar. Sows were inseminated at the beginning of their first detected estrus. Two consecutive AI sessions were performed on the first and second mornings for each sow exhibiting standing heat, utilizing disposable foam tip catheters (Minitube®, Tiefenbach, Bavaria, Germany). Each session lasted between 5 and 10 min per sow to ensure proper procedure execution and to optimize the effectiveness of the insemination.

### 2.3. Housing and General Management

The study was conducted under typical tropical conditions in northeastern Thailand, where ambient temperatures ranged from 24 to 34 °C and relative humidity ranged from 65% to 85% during the study period. The parity 1 observation was carried out during the rainy season (May–August), whereas the parity 2 observation took place during the cool season (November–January). Sows were housed in an open-housing system equipped with fans and water sprinklers to mitigate the effects of high environmental temperatures during the daytime. Each gestating sow was placed in an individual crate measuring 1.3 m^2^ (0.65 m × 2 m). Drinking water, treated with chlorine dioxide at a concentration of 2 ppm, was provided ad libitum. The sows received a gestational diet formulated according to NRC [25] standards to meet or exceed their nutritional requirements. The gestation feed consisted of a formulation based on rice bran and soybean meal containing 2900 kcal/kg metabolizable energy, 16.0% crude protein, and 0.8% lysine. On average, gestating sows were fed once daily at 8:00 AM with 2.0–2.2 kg/day from weeks 1 to 11 of gestation. From weeks 12 to 15 of gestation, the feed amount was increased to 4.0 kg/day, divided into two meals per day, while maintaining the same diet composition.

Gestating sows were transferred to the farrowing house one week before the expected date of parturition. In the farrowing house, each sow was placed in an individual crate (1.95 m × 0.75 m) positioned at the center of the farrowing pen (2.5 m × 3.0 m). The pens were fully slatted, featuring a concrete base at the center for sows and steel slats on both sides for piglets. A warm creep area (1.0 m × 6.0 m) was provided for piglets during their first week after birth. Upon relocation to the farrowing house, the sows were transitioned to a lactation diet containing 3200 kcal/kg metabolizable energy, 18.0% crude protein, and 1.0% lysine. Before parturition, they were fed 2.0 kg/day, which was reduced to 0.5–1.0 kg/day on the day of farrowing. Postpartum, the feed allowance was gradually increased by 0.5 kg/day, starting from 3.5 to 4.0 kg/day, until reaching 6.0 kg/day within one week after farrowing. During the second and third weeks of lactation, sows were provided feed ad libitum. Sows and piglets had ad libitum access to water via nipple drinkers. The lactation period lasted for 21–24 days. Boars were housed separately from the sows but within the same facility as the gestating sows. They were fed the same diet as lactating sows, receiving 2.5–3.0 kg/day in a single feeding at 8:00 AM. Water was supplied ad libitum via nipple drinkers.

The parturition process was closely monitored by stock people, ensuring minimal intervention. Assistance was provided only in cases of dystocia, defined as a birth interval exceeding 60 min between piglets. When necessary, manual fetal extraction was performed by animal husbandry specialists. Routine daily health checks were conducted by a veterinarian. Following the herd veterinarian’s recommendations, all postpartum sows received an intramuscular injection of an anti-inflammatory drug (tolfenamic acid, 40 mg/mL, 2 mg/kg, Tolfédine^®^, Vétoquinol, Lure Cedex, France) for 2 consecutive days after farrowing. Additionally, an antibiotic (amoxicillin, 1.0 mL/20 kg IM, 150 mg/mL, AMOXOIL^®^, Laboratorios Syva S.A.U., León, Spain) was administered on days 1 and 3 postpartum, and a vitamin/mineral supplement (217.8 mg/sow IM, Fercobsang^®^, Vétoquinol S.A., Lure Cedex, France) was given on day 1. Furthermore, the veterinarian recommended vaccinating all sows against foot-and-mouth disease, porcine parvovirus (Synparv MRL^®^, Syva Laboratorios, León, Spain), classical swine fever (CSF, PRO-VAC^®^, Komipharm International Co., Ltd., Siheung, Republic of Korea), Aujeszky’s disease (Novoyesky^®^, Syva Laboratorios, León, Spain), and porcine reproductive and respiratory syndrome (PRRS, Pyrsvac-183^®^, Syva Laboratorios, León, Spain).

### 2.4. Statistical Analysis

Statistical analyses were performed using SAS version 9.4 (SAS Institute, Cary, NC, USA). Descriptive statistics, including means, standard deviations, and minimum and maximum values for continuous variables, were calculated using the MEANS procedure. Categorical data, such as the incidence of cryptorchidism, were expressed as percentages.

Litter traits, including TB, BA, SB, MF, BW, and gestation length, were analyzed using the General Linear Model (GLM) procedure in SAS (PROC GLM). The statistical model included the fixed effects of sow genetic line (conventional vs. HPS), cryptorchid litter status (Crypt; yes or no), and their interaction (Genetic × Crypt). Least-squares means (LSMEANS) were obtained for each main effect and their interaction, with pairwise comparisons adjusted using Tukey’s test. Standard errors of the least-squares means and pairwise differences were also computed to assess statistical significance. Continuous data, including TB, BA, MF, SB, BW, and gestation length, were presented as least-square means ± SEM.

The association between boar type (cryptorchid vs. normal), sow genetic line (conventional vs. HPS), and the incidence of cryptorchidism in male piglets was analyzed using the Chi-square test under the FREQ procedure in SAS (PROC FREQ). Contingency tables were generated to evaluate potential differences among boars and genetic lines. The relative risk of cryptorchidism was also estimated for each combination. The incidence of cryptorchidism was expressed as a percentage. A *p* value ≤ 0.05 was considered statistically significant.

## 3. Results

### 3.1. Descriptive Data

Overall, the HPS line exhibited markedly superior prolificacy, producing approximately 33% more piglets per litter compared with the conventional line (*p* < 0.001; Table 1). Similarly, BA was higher in the HPS line compared to the conventional line (*p* < 0.001). However, piglet birth weight was lower in HPS compared to those in the conventional line (*p* < 0.001). The percentage of stillborn piglets per litter (*p* = 0.819) and the percentage of mummified fetuses per litter (*p* = 0.338) did not differ between the HPS and conventional lines (Table 2). Notably, gestation length was 0.9 days greater in the HPS line compared to the conventional line (*p* = 0.005). Cryptorchidism occurred in 25.7% (37/144) of litters inseminated with semen from the cryptorchid boar, compared to 3.8% (5/132) of litters inseminated with semen from normal boars (*p* < 0.001). In total, 42 sows produced at least one cryptorchid piglet across both parities. Among these affected sows, the average number of cryptorchid piglets per litter was 1.3 ± 0.6 (range: 1–3). In the HPS line, cryptorchidism was detected in 24.1% (26/108) of litters, whereas in the conventional line, it was observed in 30.6% (11/36) of litters (*p* = 0.441). 

### 3.2. Comparison of Reproductive Performance and Piglet Characteristics

In the HPS group, litters with cryptorchid piglets had significantly lower birth weights than those without (*p* = 0.012), whereas no significant difference was observed in the conventional line (*p* = 0.917). Litter size did not differ between litters with and without cryptorchid piglets in either line (*p* > 0.05). Number of litters with and without cryptorchid piglets in HPS and conventional genetic lines are presented in Table 3. Among the cryptorchid litters, 78.4% of sows had one cryptorchid piglet, 18.9% had two, and 2.7% had three. In contrast, when the sows were inseminated by phenotypically normal boar, all observed cryptorchid litters had only one affected piglet.

### 3.3. Incidence of Cryptorchidism in Male Piglets

The number of male piglets born from sows inseminated with semen from cryptorchid and normal boars is shown in Table 4. Cryptorchid piglets occurred far more frequently when semen from a cryptorchid boar was used, with the incidence increasing by approximately 13-fold compared with insemination using semen from normal boars (*p* < 0.001). Additionally, the majority of cryptorchid piglets (94.6%) showed unilateral cryptorchidism, while only three piglets (5.4%) exhibited bilateral cryptorchidism. Following insemination with cryptorchid boar semen, the incidence of cryptorchid piglets did not differ significantly between HPS and conventional sows. In contrast, when insemination was performed with semen from normal boars, cryptorchid piglets were rare in the HPS group and were not detected at all in the conventional group.

## 4. Discussion

### 4.1. Reproductive Performance of Conventional and HPS

In the present study, fertility, as measured by farrowing rate, was high and comparable between the two genetic groups. These findings reflect excellent insemination procedures and effective pregnancy management under commercial farm conditions [24,26]. Additionally, both groups demonstrated low rates of stillbirth and fetal mummification, resulting in a satisfactory number of piglets born alive per litter. These outcomes compare favorably with those reported in other herds raised under tropical conditions [16].

In the current study, the HPS line demonstrated superior reproductive performance, producing significantly larger litters compared to the conventional line. Likewise, the number of piglets born alive per litter was significantly greater (+3.8 BA) in the HPS line. However, this increased litter size was accompanied by lower birth weights, as piglets from the HPS line weighed significantly less than those from the conventional line. This observation aligns with our previous findings, where up to 43.3% of HPS liveborn piglets weighed less than 1.0 kg at birth [27]. Despite these differences, the percentages of stillborn piglets and mummified fetuses did not differ significantly between the two genetic lines. Notably, gestation length in the HPS line was approximately 0.9 days longer than in the conventional line. This finding contrasts with previous reports, which have consistently shown a negative correlation between gestation length and litter size, with larger litters generally associated with shorter gestation periods due to increased uterine pressure and earlier activation of parturition signals [28,29]. However, several other factors may influence gestation duration beyond litter size alone. For instance, the presence of mummified fetuses can alter intrauterine hormonal dynamics and delay the endocrine cascade required to initiate parturition, thereby prolonging gestation [30]. Moreover, gestation length is known to be a moderately heritable reproductive trait, and genetic background plays a significant role in determining variation in this parameter [31]. These genetic influences, together with the numerically higher incidence of mummified fetuses observed in the HPS group, may partially explain the extended gestation period observed in this study.

Collectively, these results suggest that although the HPS genetic line effectively enhances litter size and reproductive efficiency, additional management strategies may be necessary to mitigate challenges associated with reduced piglet birth weights and slightly prolonged gestation [32].

### 4.2. Influence of Boar Phenotype on Litter Characteristics and Cryptorchid Piglets

The current study found that cryptorchidism occurred significantly more frequently in litters inseminated with semen from cryptorchid boars than in those inseminated with semen from normal boars. However, the average number of affected piglets per litter remained relatively low, at approximately 1.3 (range: 1–3). In the first parity, the conventional line exhibited a numerically higher incidence of cryptorchidism than the HPS line, although this occurred only when semen from a cryptorchid boar was used, indicating that the paternal effect was the strongest trigger. Within the HPS group, even in the first parity, litters with low birth weights contained significantly more cryptorchid piglets, suggesting that reduced birth weight remained a contributing factor, although it was less influential than the paternal effect. In the second parity, after removing the paternal influence, low birth weight emerged as a key associated factor, reinforcing its importance in the occurrence of the condition. Fredeen and Newman [4] previously estimated the heritability of cryptorchidism to be 10.9% for the Lacombe breed and 31.4% for the Yorkshire breed. Furthermore, the association between cryptorchidism and reduced birth weight in the HPS line highlights potential management challenges, underscoring the need for careful monitoring and targeted intervention strategies, particularly in HPS herds. The incidence of cryptorchidism in piglets from sows inseminated with semen from normal boars (0.5%) was slightly lower than previously reported rates of 0.64–0.84% [33] and significantly lower than other literature values, which range from 2.2% to 6.0% [4,5,7]. This lower incidence may be attributed to the larger litter sizes observed in modern breed sows in this study. Additionally, this factor could explain the absence of differences in cryptorchid piglet incidence between the HPS and conventional groups when cryptorchid boars were used to assess the heritability of the condition.

In this study, most cryptorchid piglets exhibited unilateral cryptorchidism. This condition likely occurs because testicular descent is regulated independently on each side through the ipsilateral genitofemoral nerve (GFN) under the influence of androgens. Consequently, developmental abnormalities in the GFN or disruptions in calcitonin gene-related peptide (CGRP) signaling may contribute to the occurrence of unilateral cryptorchidism [34]. The incidence of affected sows was 25.7%, which is relatively high when using a cryptorchid boar as a positive control. Among these sows, 78.4% had a single cryptorchid piglet per litter, 18.9% had two, and 2.7% had three cryptorchid piglets. These findings indicate a genetic inheritance pattern for cryptorchidism, though its distribution among sows appears to be uneven. Despite this variation, the overall incidence remains low at approximately 6.25%, or one affected piglet per 16 piglets in litters with cryptorchid cases. In monotocous species like humans, paternal inheritance can be inferred from a small number of offspring. However, in polytocous species like pigs, where there is considerable heterogeneity in offspring growth and development [35,36], further research is needed to understand the mode of paternal inheritance of cryptorchidism. Specifically, studies should investigate piglets from affected litters that exhibit normal testicular descent.

Unexpectedly, semen from the cryptorchid boar in this study performed well, achieving an acceptable conception rate in sows. This contrasts with previous reports of low sperm counts or aspermia in cryptorchid boars [37], suggesting that some may produce sufficient sperm for limited periods. Our findings align with Frankenhuis et al. [38] and human studies [39], indicating that spermatogenic arrest in abdominally retained testes is due to elevated intra-abdominal temperature rather than congenital defects. While undescended testes can produce viable sperm until around nine months, the boar in this study remained fertile up to 13 months, suggesting a longer window of sperm production than previously thought.

### 4.3. Association Between Piglet Birth Weight and Cryptorchidism

Surprisingly, an association between cryptorchidism and lower birth weight was observed in the HPS line but not in the conventional line. Although cryptorchidism incidence did not differ between genetic lines when semen from a cryptorchid boar was used, litters with cryptorchid piglets in the HPS line had significantly lower birth weights. One possible explanation is intrauterine crowding, which is more common in HPS and may increase the risk of developmental abnormalities [18,19]. Fetal growth restriction has been associated with hormonal imbalances and impaired organ development [36], which in turn could impair testicular descent. Previous studies have clearly demonstrated that IUGR disrupts testicular development by reducing testis size, seminiferous tubule structure, and key gene expression changes [23,40]. In contrast, the conventional line, with relatively higher birth weights, showed no such association, suggesting that different genetic or environmental influences may underline cryptorchidism between the two lines. However, because we did not fully evaluate all IUGR criteria in this study, our findings should be interpreted as observational associations rather than definitive evidence of causation.

Our findings also showed no difference in total litter size between litters with and without cryptorchid piglets in either genetic line, suggesting that litter size alone does not directly contribute to cryptorchidism. Nevertheless, larger litters can lead to reduced birth weights and greater variability due to intrauterine crowding [41], which has been linked to developmental challenges [42]. In HPS litters, this variability may result in severely underdeveloped piglets, potentially increasing the risk of cryptorchidism. These results emphasize that individual birth weight, rather than total litter size, plays a more critical role, particularly in HPS lines. As selection for larger litters continues in commercial breeding, targeted strategies such as optimized sow nutrition, better fetal monitoring, and balanced genetic selection are needed to reduce low birth weight and its associated risks.

Further research should directly investigate the potential relationship between IUGR and the occurrence of cryptorchidism, particularly the underlying mechanisms by which IUGR may cause impaired testicular descent. Future studies that specifically apply morphological criteria to characterize the abnormalities associated with IUGR will be essential to clarify the role of IUGR in the occurrence of cryptorchidism. Additionally, investigations into the possible genetic and epigenetic mechanisms linking birth weight and cryptorchidism, as well as potential interventions to support fetal development in large litters, are warranted.

### 4.4. Limitations

This study had some limitations. First, the design was observational, reflecting herd circumstances rather than a randomized controlled trial. Following an African swine fever outbreak, only one phenotypically unilateral cryptorchid Duroc boar survived from a replacement group intended for culling and had reached breeding age, making him the sole sire available in the first parity. Semen from external sources was avoided due to uncertain herd health status during the epidemic and financial constraints. Although this created bias, as all first-parity sows were inseminated with the same boar, it also acted as a natural trigger that increased cryptorchid piglet numbers, thereby enhancing statistical power and allowing clearer within-group comparisons when normal boars were used in the second parity. Second, the sequential design prevented a balanced 2 × 2 factorial analysis, leaving potential contemporary group effects uncontrolled. Third, intrauterine growth restriction was not directly measured, so the observed association between birth weight and cryptorchidism should be regarded as an observational correlation rather than proof of causation. Another limitation of this study is that only three normal boars, all full siblings, were used for insemination in parity 2. This restricted genetic variability in the paternal line, which may limit the external validity of our findings and should be considered when extrapolating the results to broader pig populations.

## 5. Conclusions

The present study confirmed the heritability of cryptorchidism through the use of a cryptorchid boar phenotype. The significantly lower average birth weight observed in cryptorchid litters within the HPS line, but not in conventional sow lines, suggests that modern hyperprolific breeds may influence the incidence of cryptorchidism in commercial pig farms. Low birth weight, commonly associated with intrauterine growth restriction (IUGR), may contribute to testicular descent failure; however, further studies are required to confirm this relationship. These findings emphasize the importance of optimizing fetal growth conditions, including improved gestational nutrition and fetal monitoring, to reduce cryptorchidism risk in HPS populations.

## Figures and Tables

**Table 1 animals-15-03105-t001:** Number of piglets across sow genetic lines, parity groups, and boar types.

Parity Groups/ Boar Type	Sow	Litters	Piglets	Males	Females
First parity/ Cryptorchid boar	HPS	108	1441	673 *	768
	Conventional	36	385	203	182
Second parity/ Normal boar	HPS	100	1771	896	875
	Conventional	32	406	217	189
Total		276	4003	1989	2014

HPS = hyperprolific sow; * One hermaphroditic piglet was included in descriptive counts but was excluded from cryptorchidism analyses.

**Table 2 animals-15-03105-t002:** Reproductive data of hyperprolific sows (HPS) compared with conventional sows (least square means ± SEM).

Variables	Genetic Lines	
	Conventional	HPS
Number of litters	68	208
Farrowing rate (%)	89.5	90.8
Gestation length (days)	115.6 ± 0.3 ^a^	116.5 ± 0.2 ^b^
Total number of piglets born per litter	12.3 ± 0.8 ^a^	16.5 ± 0.5 ^b^
Number of piglets born alive per litter	11.6 ± 0.8 ^a^	15.4 ± 0.5 ^b^
Stillborn piglets (%)	3.3 ± 1.3	3.7 ± 0.8
Mummified fetuses (%)	1.9 ± 1.6	3.7 ± 0.9
Piglets birthweight (kg)	1.30 ± 0.02 ^a^	1.14 ± 0.01 ^b^

^a,b^ Different superscripts indicated significant difference between groups (*p* ≤ 0.05).

**Table 3 animals-15-03105-t003:** Comparison of reproductive performance and piglet characteristics between hyperprolific sows (HPS) and conventional sows, categorized by presence of cryptorchidism (least square means ± SEM).

Variables	Conventional		HPS	
	Normal	Cryptorchid	Normal	Cryptorchid
Number of litters	57	11	177	31
Gestation length (days)	116.1 ± 0.2 ^ab^	115.1 ± 0.5 ^a^	116.3 ± 0.1 ^ab^	116.7 ± 0.3 ^b^
Total number of piglets born per litter	12.4 ± 0.6 ^a^	12.3 ± 1.5 ^a^	16.6 ± 0.4 ^b^	16.4 ± 0.9 ^b^
Number of piglets born alive per litter	11.6 ± 0.6 ^a^	11.6 ± 1.4 ^a^	15.4 ± 0.4 ^b^	15.4 ± 0.9 ^b^
Stillborn piglets (%)	4.02 ± 1.02 ^a^	2.64 ± 2.32 ^a^	3.68 ± 0.58 ^a^	3.65 ± 1.38 ^a^
Mummified fetuses (%)	1.74 ± 1.30 ^a^	2.00 ± 2.96 ^a^	2.79 ± 0.74 ^a^	4.55 ± 1.76 ^a^
Piglets birthweight (kg)	1.28 ± 0.02 ^a^	1.31 ± 0.04 ^a^	1.18 ± 0.01 ^b^	1.11 ± 0.02 ^c^

^a,b,c^ Different superscripts indicated significant difference between groups (*p* ≤ 0.05).

**Table 4 animals-15-03105-t004:** Incidence of cryptorchidism in male piglets of hyperprolific sows (HPS) and conventional sows inseminated with semen from cryptorchid and normal boars.

Boar Phenotype	Sow	Normal Male Piglets	Cryptorchid Piglets	Percentage
Cryptorchid boar	HPS	635	37	5.8
	Conventional	189	14	7.4
	Total	824	51	6.2
Normal boar	HPS	891	5	0.6
	Conventional	217	0	0
	Total	1108	5	0.5

## Data Availability

The original contributions presented in this study are included in the article. Further inquiries can be directed to the corresponding author(s).

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
