# Peer review of "Genetic Inheritance and the Impact of Low Birth Weight on the Incidence of Cryptorchidism in Hyperprolific Sows"

_animals, 2025, doi:10.3390/ani15213105_

Round 1

Reviewer 1 Report

Comments and Suggestions for Authors

The manuscript “Genetic inheritance and the impact of low birth weight on the incidence of cryptorchidism in hyperprolific sows” is well written manuscript. However, it has some shortcomings

Generally there is inconsistency in use of abbreviations and repetition of the text. Authors are advised to check all manuscript for these important points. Here are the specific comments

Simple summary needs to be re-written especially the lines 15-19 looks like materials and methods explanation of the study which is not needed. Focus non technical readers and write it again by simplifying

Line 44: Why not abbreviation for hyperprolific sows

Introduction looks to have less necessary material discussed more than the essential one like genetic inheritance, birth weight, hyperprolific sows. Authors are suggested to do the necessary improvements

60-63: This is not relevant to the study, if you still want to use you can use it very briefly.

Why authors prefer to abbreviate hyperprolific sow at line number 91 and not earlier at line number 88?

99: bringing humans into the text not needed

119: whay not abbreiviation here for hyperprolific sow. Please check all manuscript for this, there are other places which needs attention like line 349

147: ml or mL. Check at all places

166: Soybean or soybean meal?

210: Do these terms need to be explained before use as abbreviations

Line 226: As this information, 16.5 ± 0.3 vs. 12.4 ± 0.6, is as such in the table no need to write in the text. Authors are advised not to repeat the information in text and tables. Tables should provide distinct information. It is strongly suggested to check all manuscript for such inconsistencies. Authors if want to present try another way like xxx% higher was observed, something like this. Another suggestion is to present some information as figure instead of table, e.g., table 3. In this way text can have more room to present values and figure bar graph as a visual.

Table 3: 51 Cryptorchid piglets out of total 824 is not 5.8%. Authors are requested to correct it and verify all other values

Line 273: 90.8% for HPS sows and 89.5%. Authors are strongly encouraged to avoid the use of such values in discussion which already have been used in Results part. Please correct all discussion for such redundency

Same at line 277

Discussion is mostly the repetition of the results. Please compare your findings with previous literature and discuss why? Authors are suggested to reconstruct the discussion.

Conclusion: Conclusion should contain the key message

Author Response

We would like to express our sincere gratitude to Reviewer 1 for the thorough review and insightful comments. We have carefully addressed each point raised, and the manuscript has been substantially improved in response.

Detailed point-by-point responses and explanations are provided below.

Comments 1: Simple summary needs to be re-written especially the lines 15–19 looks like materials and methods explanation of the study which is not needed. Focus non-technical readers and write it again by simplifying.

Response: We have carefully revised the simple summary to improve readability. The revised version removes methodological details (previously in lines 15–19)

Comments 2: Line 44: Why not abbreviation for hyperprolific sows

Response: The term hyperprolific sows has now been abbreviated as HPS (Line 43). This word and its abbreviation were carefully examined and consistently used throughout the manuscript.

Comments 3: Introduction looks to have less necessary material discussed more than the essential one like genetic inheritance, birth weight, hyperprolific sows. Authors are suggested to do the necessary improvements

Response: We appreciate the reviewer’s insightful suggestion. The introduction section has been carefully revised. Specifically, we have expanded the introduction of genetic inheritance at line number 59-61, piglet birth weight and hyperprolific sow characteristics at line number 94-100

Comments 4: 60-63: This is not relevant to the study, if you still want to use you can use it very briefly.

Response: In accordance with the comment, we have revised the sentence in lines 61–62 to present the information more briefly.

The revised text now reads: Practical control strategies include selecting sire lines free of cryptorchidism and excluding affected animals. However, the condition still occurs [4].”

Comments 5: Why authors prefer to abbreviate hyperprolific sow at line number 91 and not earlier at line number 88?

Response: The abbreviation “HPS” has now been introduced at its first occurrence (line 87) and is used consistently throughout the manuscript.

Comments 6: 99: bringing humans into the text not needed

Response: We have revised the text by removing the reference to “humans” in this sentence.

The revised version now reads: “Given the established risk factors for cryptorchidism in pigs. (line 103)

Comments 7: 119: why not abbreiviation here for hyperprolific sow. Please check all manuscript for this, there are other places which needs attention like line 349

Response: We have carefully reviewed the entire manuscript for consistency in abbreviation usage and have now revised all instances to ensure that hyperprolific sow (HPS) is consistently abbreviated throughout the text.

Comments 8: 147: ml or mL. Check at all places

Response: We have chosen to use mL (line 173) as the standard unit abbreviation and have carefully checked and revised all occurrences throughout the manuscript for consistency.

Comments 9: 166: Soybean or soybean meal?

Response: Thank you for pointing this out. We confirm that the correct term is soybean meal (line 197), and the manuscript has been revised accordingly.

Comments 10: 210: Do these terms need to be explained before use as abbreviations

Response: These abbreviations TB (total born piglets), BA (born alive), SB (stillborn), MF (mummified fetuses), and BW (birth weight) are already defined at their first appearance in the Materials and Methods section (line 150-152) (Section 2.1, Study Design, before the final paragraph).

Comments 11: Line 226: As this information, 16.5 ± 0.3 vs. 12.4 ± 0.6, is as such in the table no need to write in the text. Authors are advised not to repeat the information in text and tables. Tables should provide distinct information. It is strongly suggested to check all manuscript for such inconsistencies. Authors if want to present try another way like xxx% higher was observed, something like this.

Response: We have carefully revised the Results section to avoid unnecessary repetition of data already presented in the tables. Instead of repeating the same numerical values, we now present the findings in a comparative and descriptive manner at line 258-259, line 278-281, line 291-294 and line 297-301. It now reads:

  • line 258-259
  • ‘’ Overall, the HPS line exhibited markedly superior prolificacy, producing ap-proximately 33% more piglets per litter compared with the conventional line (p < 0.001; Table 1).’’
  • 278-281
  • ‘’In the HPS group, litters with cryptorchid piglets had significantly lower birth weights than those without (p = 0.012), whereas no significant difference was observed in the conventional line (p = 0.917). Litter size did not differ between litters with and without cryptorchid piglets in either line (p > 0.05).’’
  • line 291-294
  • ‘’ The number of male piglets born from sows inseminated with semen from cryptorchid and normal boars is shown in Table 4. Cryptorchid piglets occurred far more frequently when semen from a cryptorchid boar was used, with the incidence increasing by approximately 13-fold compared with insemination using semen from normal boars (p < 0.001).’’

297-301

‘’ the incidence of cryptorchid piglets did not differ significantly between HPS and con-ventional sows when semen from a cryptorchid boar was used. In contrast, when in-semination was performed with semen from normal boars, cryptorchid piglets were rare in the HPS group and were not detected at all in the conventional group.’’

Comments 12: Another suggestion is to present some information as figure instead of table, e.g., table 3. In this way text can have more room to present values and figure bar graph as a visual.

Response: We appreciate the reviewer’s suggestion. We initially attempted to visualize the data from Table 3 using a bar graph; however, because the values differ greatly in magnitude (single-digit values versus several hundred), the resulting figure was visually distorted and did not clearly represent the data. For this reason, we believe that presenting the data in table is the most effective and accurate way to convey the information.

Comments 13: Table 3: 51 Cryptorchid piglets out of total 824 is not 5.8%. Authors are requested to correct it and verify all other values

Response: We appreciate the reviewer’s careful reading and the opportunity to correct this mistake. We have corrected the value in Table 4 (a new Table 1 has been added in the Materials and Methods section as suggested by Reviewer 4. Consequently, all subsequent tables in the Results section have been renumbered): 51/824 male piglets = 6.2%, not 5.8%. The corresponding text in the Results section has been updated accordingly, and we rechecked all counts, denominators, and percentages in Table 4 and related sentences for consistency (rounding now reported to one decimal place).

Comments 14: Line 273: 90.8% for HPS sows and 89.5%. Authors are strongly encouraged to avoid the use of such values in discussion which already have been used in Results part. Please correct all discussion for such redundancy

Response: We have revised the Discussion section accordingly. The sentence previously including redundant numerical values has been rephrased to remove repetition from the Results section. It now reads (line 307-309):

“In the present study, fertility, as measured by farrowing rate, was high and comparable between the two genetic groups. These findings reflect excellent insemination procedures and effective pregnancy management under commercial farm conditions [24,26].”

Comments 15: Same at line 277

Response: We have also revised the sentence at line 309-312 to remove repeated numerical values and focus on interpretation rather than restating results. It now reads:

“Additionally, both groups demonstrated low rates of stillbirth and fetal mummification, resulting in a satisfactory number of piglets born alive per litter. These outcomes compare favorably with those reported in other herds raised under tropical conditions [16].”

Comments 16: Discussion is mostly the repetition of the results. Please compare your findings with previous literature and discuss why? Authors are suggested to reconstruct the discussion.

Response: Thank you for your comment. We have gone through the Discussion section and given more discussion (compare findings with previous literature and discuss why) at line 322-333, line 342-350 and line 394-398. It now reads:

line 322-333

“This finding contrasts with previous reports, which have consistently shown a negative correlation between gestation length and litter size, with larger litters generally associated with shorter gestation periods due to increased uterine pressure and earlier activation of parturition signals [28,29]. However, several other factors may influence gestation duration beyond litter size alone. For instance, the presence of mummified fetuses can alter intrauterine hormonal dynamics and delay the endocrine cascade re-quired to initiate parturition, thereby prolonging gestation [30]. Moreover, gestation length is known to be a moderately heritable reproductive trait, and genetic background plays a significant role in determining variation in this parameter [31]. These genetic influences, together with the numerically higher incidence of mummified fetuses observed in the HPS group, may partially explain the extended gestation period observed in this study.”

line 342-350

“In the first parity, the conventional line exhibited a numerically higher incidence of cryptorchidism than the HPS line, although this occurred only when semen from a cryptorchid boar was used, indicating that the paternal effect was the strongest trigger. Within the HPS group, even in the first parity, litters with low birth weights contained significantly more cryptorchid piglets, suggesting that reduced birth weight remained a contributing factor, although it was less influential than the paternal effect. In the second parity, after removing the paternal influence, low birth weight emerged as a key associated factor, reinforcing its importance in the occurrence of the condition.”

line 394-398

“Fetal growth restriction has been associated with hormonal imbalances and impaired organ development [36], which in turn could impair testicular descent. Previous studies have clearly demonstrated that IUGR disrupts testicular development by reducing testis size, seminiferous tubule structure, and key gene expression changes [23,40].’’

We would like to note that we have already updated all relevant references

Comments 17: Conclusion: Conclusion should contain the key message

Response: We have revised the Conclusion section (line 440-448) to better highlight the key message of our study. The updated conclusion now focuses on the main findings and their implications, as follows:

‘’The present study confirmed the heritability of cryptorchidism through the use of a cryptorchid boar phenotype. The significantly lower average birth weight observed in cryptorchid litters within the HPS line, but not in conventional sow lines, suggests that modern hyperprolific breeds may influence the incidence of cryptorchidism in commercial pig farms. Low birth weight, commonly associated with intrauterine growth restriction (IUGR), may contribute to testicular descent failure; however, further studies are required to confirm this relationship. These findings emphasize the importance of optimizing fetal growth conditions, including improved gestational nutrition and fetal monitoring, to reduce cryptorchidism risk in HPS populations.’’

We sincerely appreciate Reviewer 1’s valuable comments and constructive suggestions, which have significantly strengthened the scientific quality and presentation of our manuscript. We are especially thankful for the opportunity to identify and correct several mistakes, allowing us to substantially improve the clarity and rigor of this work.

Reviewer 2 Report

Comments and Suggestions for Authors

This manuscript describes experiments to compare hyperprolific sows (HPS) with conventional strains and examine the incidence of cryptorchidism in piglets. The authors found no significant difference in the incidence of cryptorchidism. However, they demonstrated that piglets from HPS sows had lower birth weights, which were significantly associated with the incidence of cryptorchidism. While there were some numerical differences in cryptorchidism rates between the conventional and HPS lines, these differences were not statistically significant (lines 296-298). The discussion converges on “the association with low birth weight,” giving an undeniable impression that the focus is misplaced. The reviewer recommended restructuring the paper to include the discovery that low birth weight is a risk factor.

This observational study utilized large-scale data and was conducted at actual commercial pig farms, suggesting high field applicability.

Author Response

We sincerely thank Reviewer 2 for their thoughtful evaluation and constructive comments on our manuscript. We highly appreciate the reviewer’s valuable suggestions, particularly regarding the need to restructure the discussion and better emphasize the role of low birth weight as a potential risk factor for cryptorchidism. In response, we have carefully revised the Discussion and Conclusion sections to reflect this key point more clearly and to ensure that the manuscript highlights the significance of birth weight within the context of our findings. Our detailed point-by-point responses are provided below.

Comments : While there were some numerical differences in cryptorchidism rates between the conventional and HPS lines, these differences were not statistically significant (lines 296-298). The discussion converges on “the association with low birth weight,” giving an undeniable impression that the focus is misplaced. The reviewer recommended restructuring the paper to include the discovery that low birth weight is a risk factor.

Response: We agree with the reviewer. Accordingly, we have carefully revised the relevant section to improve the clarity and logical flow of our interpretation (line 342-350).

It now reads:

“In the first parity, the conventional line exhibited a numerically higher incidence of cryptorchidism than the HPS line, although this occurred only when semen from a cryptorchid boar was used, indicating that the paternal effect was the strongest trigger. Within the HPS group, even in the first parity, litters with low birth weights contained significantly more cryptorchid piglets, suggesting that reduced birth weight remained a contributing factor, although it was less influential than the paternal effect. In the second parity, after removing the paternal influence, low birth weight emerged as a key associated factor, reinforcing its importance in the occurrence of the condition.

We sincerely thank Reviewer 2 for the constructive comments and valuable suggestions, which helped us improve the clarity, focus, and overall quality of our discussion. Your feedback has been instrumental in strengthening the manuscript and ensuring that the key findings are more clearly presented.

Reviewer 3 Report

Comments and Suggestions for Authors

The article submitted for review addresses a very interesting and timely topic regarding cryptorchidism in hyperprolific sows. The article is coherent and well-written, but after reading it, several comments arise.

I consider the "Limitations" section to be positive, explaining why only one cryptorchid boar was used to inseminate the gilts; however, why was semen from three boars used to inseminate the sows in their second heat?  Did the Authors consider using the semen of only one boar in this situation? Additionally, the abbreviation ASF (line 376) appears in this section, the full text of which was not previously included in the article.

The "Materials and Methods" section lacks information on the number of sows (conventional vs. HPS), as only the number of litters is provided. Furthermore, 144 gilts were used in the study, while 132 sows gave birth in the second litter, meaning 12 sows were culled after the first litter. Unfortunately, there is no information about this, nor about the reasons for culling. At this point, however, I would like to emphasize the very detailed description of subsections 2.2 (Semen preparation and insemination techniques) and 2.3 (Housing and general management).

The paper uses simple descriptive statistics and the GLM method. The authors write: "The statistical models included the effects of genetic lineage, cryptorchid litter status (yes or no), and their interaction." Therefore, in my opinion, it is absolutely essential to present this model(s) along with an explanation!

Authors should definitely avoid sentences like "Table 1 shows the descriptive statistics on sow reproductive performances and farrowing characteristics" (lines 224-225), which have no substantive content. It's definitely better to write: "On average, the HPS line produced more piglets per litter than the conventional line 16.5 ± 0.3 vs. 12.4 ± 0.6, p < 0.001 (Tab. 1)".

Similarly, lines 245-247: if Authors absolutely do not want to avoid such sentences, they should be placed at the beginning of the paragraph, not in the middle, after the description of the first results! Consequently, I also recommend removing the sentence from lines 255-266.

Both in the Abstract (line 35) and the Results section (line 233) the phrase "Among affected sows (n = 42)..." appears - however, unfortunately, I don't understand what this means, what kind of sows these are, and why 42? I couldn't find an explanation for selecting this group, its description or mentioning it in the article.

The results presented in the tables also raise concerns. First, knowing that the study only included sows after their first and second farrowings, I'm not sure whether the "Parity number" variable is necessary (Tab. 1 and 2).   Furthermore, if the Table 1 provides a p-value (and only two groups are compared), I'm not sure if it's appropriate to include superscript letters (a, b) indicating significant differences. However, I consider it completely unjustified to include the same letters (a, a) when there are no significant differences.   The data in Table 2 also require a major correction. First, the "Farrowing rate" cannot exceed 100% (if we understand it as the number of farrowings relative to inseminated females). The gestation length is also absolutely impossible (!); it looks a bit like the Authors mixed up the rows and entered the data on "Gestation length" into the row regarding the "Farrowing rate"!   The data regarding the total number of piglets born per litter (TB) and the number of piglets born alive per litter (BA) also raise doubts. First, the Authors obtained identical results for both variables, which is unlikely. Additionally, in the Material and Methods section, the Authors indicated that the HPS sows selected for the study gave birth to more than 16 piglets per litter, so the TB number should have been above 16.   Such errors should not appear in scientific articles, so special attention is recommended!   It is worth adding the letter "c" as a superscript under Table 2, as piglets from HPS "cryptorchid" sows differ from all others.   Have the authors considered presenting the results in Table 3 separately for gilts and sows?   The discussion raises no objections.

Author Response

Thank you very much for your positive feedback and constructive suggestions. We appreciate your recognition of the “Limitations” section and the rationale behind using a single cryptorchid boar for the first parity, which we aimed to clearly explain in the manuscript. We have carefully reviewed and revised the manuscript in response to all subsequent comments you provided, as detailed below.

Comments 1: Why was semen from three boars used to inseminate the sows in their second heat?  Did the Authors consider using the semen of only one boar in this situation?

Response: Thank you for your comment. Due to the African swine fever outbreak, one cryptorchid boar was used for the first parity, as mentioned in the Limitations section. However, using a single boar as the treatment stimulator likely helped minimize variation related to boar effects. However, during that time, only small batches of 10–15 sows could be inseminated at a time, requiring an extended period to complete insemination for all sows. In the second parity, the farm owner aimed to improve the insemination schedule and efficiency. Therefore, semen from three normal boars that were full siblings were used to minimize variation between boars as much as possible.

Comments 2: The abbreviation ASF (line 376) appears in this section, the full text of which was not previously included in the article.

Response: We have corrected the manuscript by writing out the full term “African swine fever” (line 423) instead of using the abbreviation ASF in that section.

Comments 3: The "Materials and Methods" section lacks information on the number of sows (conventional vs. HPS), as only the number of litters is provided.

Response: We appreciate the reviewer’s comment and have added detailed information on the number of sows from each genetic line in the Materials and Methods section (line 119-122).

It now reads:

“A total of 144 Landrace × Yorkshire crossbred sows were included in the study, consisting of 108 HPS and 36 conventional sows in the first parity. All 108 HPS and 36 conventional gilts were selected from the entire sow population on the farm during the study period based on the reproductive performance records of their parental lines.

Comments 4: 144 gilts were used in the study, while 132 sows gave birth in the second litter, meaning 12 sows were culled after the first litter. Unfortunately, there is no information about this, nor about the reasons for culling.

Response: We have now clarified the number of sows that did not produce a second litter and provided the reasons for their removal from the study (line 122-126).

It now reads:

‘’However, during weaning to the next farrowing, 12 sows were culled due to reproductive failures, including absence of estrus, abortion, complete litter mummification, and sudden death. Therefore, 100 HPS and 32 conventional sows which gave birth in the second parity were used.’’

Comments 5: The paper uses simple descriptive statistics and the GLM method. The authors write: "The statistical models included the effects of genetic lineage, cryptorchid litter status (yes or no), and their interaction." Therefore, in my opinion, it is absolutely essential to present this model(s) along with an explanation!

Response: We appreciate the reviewer’s valuable comment and fully agree that the statistical model and its explanation should be clearly described. In response, we have revised the Materials and Methods section (line 240-246 and line 249-255) to provide a more detailed explanation of the analytical approach and the model used. The revised text now reads as follows:

 line 240-246

“Litter traits, including TB, BA, SB, MF, and BW and gestation length were analyzed using the General Linear Model (GLM) procedure in SAS (PROC GLM). The statistical model included the fixed effects of sow genetic line (conventional vs. HPS), cryptorchid litter status (Crypt; yes or no), and their interaction (Genetic × Crypt). Least-squares means (LSMEANS) were obtained for each main effect and their interaction, with pairwise comparisons adjusted using Tukey’s test. Standard errors of the least-squares means and pairwise differences were also computed to assess statistical significance.”

line 249-255

“The association between boar type (cryptorchid vs. normal), sow genetic line (conventional vs. HPS), and the incidence of cryptorchidism in male piglets was analyzed using the Chi-square test under the FREQ procedure in SAS (PROC FREQ). Contingency tables were generated to evaluate potential differences among boars and ge-netic lines. The relative risk of cryptorchidism was also estimated for each combination. The incidence of cryptorchidism was expressed as a percentage. A p value ≤ 0.05 was considered statistically significant.”

Comments 6: Consequently, I also recommend removing the sentence from lines 255-266.

Response: We thank the reviewer for this comment. Rather than removing the section entirely, we chose to rephrase it for conciseness because this part is also linked to comments from other reviewers who requested us to minimize redundancy while still keeping essential information. In the revised manuscript (Lines 291-294), we have restructured the paragraph to avoid unnecessary repetition of the data already presented in Table 4 while retaining its relevance.

Additionally, we would like to note that some details described in this paragraph are not included in the table (e.g., the proportion of unilateral vs. bilateral cryptorchidism and the occurrence patterns across genetic lines under different insemination conditions).

Comments 7: Both in the Abstract (line 35) and the Results section (line 233) the phrase "Among affected sows (n = 42)..." appears - however, unfortunately, I don't understand what this means, what kind of sows these are, and why 42? I couldn't find an explanation for selecting this group, its description or mentioning it in the article.

Response: To improve clarity, we have added a brief explanation in the Abstract (line 32-33) and Results section (line 267-268). The sentence now reads:

‘’In total, 42 sows produced at least one cryptorchid piglet across both parities.’’

Comments 8: The results presented in the tables also raise concerns. First, knowing that the study only included sows after their first and second farrowing, I'm not sure whether the "Parity number" variable is necessary (Tab. 1 and 2).

Response: We appreciate the reviewer’s insightful comment. In response, we have removed the “Parity number” variable from Tables 2 and 3 (a new Table 1 has been added in the Materials and Methods section as suggested by Reviewer 4. Consequently, all subsequent tables in the Results section have been renumbered).

Comments 9: if the Table 1 provides a p-value (and only two groups are compared), I'm not sure if it's appropriate to include superscript letters (a, b) indicating significant differences. However, I consider it completely unjustified to include the same letters (a, a) when there are no significant differences.

Response: In response, we have removed the redundant superscript letters (e.g., a, a) from Table 2 to avoid potential confusion, and we have also removed the separate p-value column to simplify data presentation.

Comments 10: The data in Table 2 also require a major correction. First, the "Farrowing rate" cannot exceed 100% (if we understand it as the number of farrowings relative to inseminated females). The gestation length is also absolutely impossible (!); it looks a bit like the Authors mixed up the rows and entered the data on "Gestation length" into the row regarding the "Farrowing rate"!   The data regarding the total number of piglets born per litter (TB) and the number of piglets born alive per litter (BA) also raise doubts. First, the Authors obtained identical results for both variables, which is unlikely.

Response: We sincerely thank the reviewer for carefully pointing out these important issues and deeply apologize for the oversight in the previous version. After rechecking the original dataset, we identified that the “Farrowing rate” and “Gestation length” rows were mistakenly swapped during table preparation. In addition, the values for total piglets born (TB) and piglets born alive (BA) were incorrectly reported and have now been thoroughly reverified and corrected.

Comments 11: Additionally, in the Material and Methods section, the Authors indicated that the HPS sows selected for the study gave birth to more than 16 piglets per litter, so the TB number should have been above 16.   Such errors should not appear in scientific articles, so special attention is recommended!

Response: We deeply apologize for the mistake. Upon reexamining our dataset and original records, we discovered that this issue resulted from an error in data entry during manuscript preparation, which subsequently led to an incorrect interpretation of the results.

We have now carefully rechecked the original data and corrected the total born (TB) and born alive (BA) values in Table 3 to accurately reflect the actual outcomes. The revised values are now consistent with the expected prolificacy of HPS sows (>16 total born piglets per litter).

We are grateful for the reviewer’s comment, which allowed us to identify and correct this mistake

Comments 12: It is worth adding the letter "c" as a superscript under Table 2, as piglets from HPS "cryptorchid" sows differ from all others.

Response: We thank the reviewer for this helpful suggestion. we included a note under the table stating (line 289)

‘’a,b,c Different superscripts indicate significant differences between groups (p ≤ 0.05).’’

Comments 13: Have the authors considered presenting the results in Table 3 separately for gilts and sows?

Response:

We thank the reviewer for this valuable suggestion. We carefully considered presenting the results in Table 4 separately for gilts and sows; however, we believe this approach would not provide additional information. This is because all sows in the first parity were inseminated with semen from the cryptorchid boar, while in the second parity all sows were inseminated with semen from normal boars. As a result, the influence of parity is linked to the boar phenotype used, and the difference between gilts and sows can already be interpreted based on the insemination group rather than by separating the data in the table.

For this reason, we prefer to retain the original table format, which we believe provides the clearest and most relevant presentation of the results in line with the objectives of the study. We hope the reviewer understands our rationale for this decision.

We sincerely thank the reviewer for their positive and constructive feedback throughout the review process. We are pleased that the discussion raised no objections and greatly appreciate the reviewer’s time and thoughtful evaluation, which helped us improve the overall quality and clarity of the manuscript.

Reviewer 4 Report

Comments and Suggestions for Authors

This manuscript investigated the genetic inheritance of cryptorchidism in pigs and its association with low birth weight in hyperprolific sows. The dataset is large (> 4000 piglets) and the results provide new insights into the role of birth weight in cryptorchidism risk.

my comments:

Line 43-44: the authors imply causality (low birth weight - cryptorchidism) rather than emphasizing the observational nature of the findings. but in Table 2: this was only true in the HPS sow group. In the Conventional sow group, there was no association between piglet birth weight and cryptorchidism.

Line 117: The total piglets are reported (n=4003), but distribution across genetic lines, parities, and boar types could be tabulated in the methods rather than only in the results.

Line 119: The definition of hyperprolific sows (>16 piglets) is mentioned, but the selection of the 208 HPS litters vs. 68 conventional litters could be explained more clearly.

Line 126: Only three normal boars (full siblings) were used in parity 2. This limits genetic variability in the paternal line and may reduce external validity of findings. This should be stated in the Discussion.

Line 135: Who performed the palpation (veterinarian or farm staff)? Was the diagnosis confirmed later (at weaning) to avoid false negatives? Inter-observer consistency (was palpation checked by more than one person)?

Line 160: Housing, feeding, and vaccination are described in detail. However, environmental conditions (temperature, humidity, seasonality) during the study period could also affect reproductive performance. please add these information.

Line 369: The discussion of intrauterine growth restriction (IUGR) is speculative since IUGR was not measured, and should be toned down or as a hypothesis for future research.

Line 393: The conclusion that “birth weight plays a critical role” is somewhat overstated given the design.

Table 2: the third line "Farrowing rate (%)" should be "Gestation length (days)" and the fifth and sixth line data is the same. please check it carefully.

Author Response

We sincerely thank Reviewer 4 for the thorough evaluation of our manuscript and the positive assessment of our work. We appreciate the recognition of the dataset size and the value of our findings, as well as the constructive suggestions provided to further improve the clarity, methodological transparency, and scientific rigor of the study. We have carefully addressed each comment, and detailed point-by-point responses are provided below.

Comments 1: Line 43-44: the authors imply causality (low birth weight - cryptorchidism) rather than emphasizing the observational nature of the findings. but in Table 2: this was only true in the HPS sow group. In the Conventional sow group, there was no association between piglet birth weight and cryptorchidism.

Response: We agree that the original wording in the Abstract could misleadingly suggest a causal relationship. To address this, we have revised the sentence to clearly emphasize that the relationship identified in this study is observational rather than causal (line 40-43). We also now specify that the association between low birth weight and cryptorchidism was observed only in the HPS group. The sentence now reads:

‘’In conclusion, the lower average birth weight in cryptorchid litters of the HPS line, but not in conventional lines, suggests that HPS breeds may influence cryptorchidism incidence. These findings highlight the need to optimize fetal growth especially in the HPS to reduce this risk’’

Comments 2: Line 117: The total piglets are reported (n=4003), but distribution across genetic lines, parities, and boar types could be tabulated in the methods rather than only in the results.

Response: Thank you for this valuable suggestion. In response, we have added a new table (Table 1) in the Materials and Methods section to clearly present the number of piglets by genetic line, parity, and boar type.

Comments 3: Line 119: The definition of hyperprolific sows (>16 piglets) is mentioned, but the selection of the 208 HPS litters vs. 68 conventional litters could be explained more clearly.

Response: We have now clarified the selection criteria for the HPS and conventional groups in the Materials and Methods section (line 119-126). The sentence now reads:

‘’A total of 144 Landrace × Yorkshire crossbred sows were included in the study, consisting of 108 HPS and 36 conventional sows in the first parity. All 108 HPS and 36 conventional gilts were selected from the entire sow population on the farm during the study period based on the reproductive performance records of their parental lines. However, during weaning to the next farrowing, 12 sows were culled due to reproductive failures, including absence of estrus, abortion, complete litter mummification, and sudden death. Therefore, 100 HPS and 32 conventional sows which gave birth in the second parity were used. ‘’

Comments 4: Line 126: Only three normal boars (full siblings) were used in parity 2. This limits genetic variability in the paternal line and may reduce external validity of findings. This should be stated in the Discussion.

Response: We agree with the reviewer’s observation. In response, we have added a statement in the Discussion (line 435-438) (Section 4.4, Limitations) to acknowledge this limitation.

Comments 5: Line 135: Who performed the palpation (veterinarian or farm staff)? Was the diagnosis confirmed later (at weaning) to avoid false negatives? Inter-observer consistency (was palpation checked by more than one person)?

Response: We have revised the Materials and Methods section to provide a more detailed description of the diagnostic procedure (line 155-164). Specifically, we now state that

‘’the presence of cryptorchidism was evaluated by palpation between days 1 and 7 postpartum. All male piglets were initially examined by trained farm staff under the supervision of a licensed veterinarian within 24 hours after birth. Each palpation was immediately re-checked by the veterinarian during the examination to ensure diagnostic accuracy. Piglets in which one or both testes were absent from the scrotum were classified as cryptorchid.

To minimize the risk of false negatives, a second examination was routinely performed at weaning (28 days of age), during which cryptorchid piglets underwent surgical removal of undescended testes if the condition was confirmed. The percentage of cryptorchid piglets was calculated based on the total number of male piglets within each litter and expressed as a percentage.’’

Comments 6: Line 160: Housing, feeding, and vaccination are described in detail. However, environmental conditions (temperature, humidity, seasonality) during the study period could also affect reproductive performance. please add these information.

Response: We have revised the Materials and Methods section to include detailed information about the environmental conditions during the study period (line 187-191).

The sentence now reads:

‘’The study was conducted under typical tropical conditions in northeastern Thailand, where ambient temperatures ranged from 24 to 34 °C and relative humidity ranged from 65% to 85% during the study period. The parity 1 observation was carried out during the rainy season (May–August), whereas the parity 2 observation took place during the cool season (November–January).’’

Comments 7: Line 369: The discussion of intrauterine growth restriction (IUGR) is speculative since IUGR was not measured, and should be toned down or as a hypothesis for future research.

Response: Thank you for this valuable comment. We agree with the reviewer’s observation. In response, we have revised the relevant part of the Discussion to make it clear that IUGR was not directly measured in this study at line 400-402 and line 413-417 and that any reference to it should be interpreted as a possible explanation or hypothesis rather than a confirmed mechanism.

Comments 8: Line 393: The conclusion that “birth weight plays a critical role” is somewhat overstated given the design.

Response: We agree with the reviewer that the original wording may have overstated the strength of our conclusion. To address this, we have revised the sentence (line 441-444) to adopt a more cautious and accurate tone that reflects the observational nature of our study. The revised version now states:

“The significantly lower average birth weight observed in cryptorchid litters within the HPS line, but not in the conventional sow lines, implied the potential influence of modern hyperprolific sow breeds on the incidence of cryptorchidism in commercial pig farms.”

Comments 9: Table 2: the third line "Farrowing rate (%)" should be "Gestation length (days)" and the fifth and sixth line data is the same. please check it carefully.

Response: Thank you very much for pointing out this mistake. We carefully rechecked Table 3 (a new Table 1 has been added in the Materials and Methods section as suggested by Reviewer 4. Consequently, all subsequent tables in the Results section have been renumbered) and confirmed that the third row was mislabeled. It should indeed be “Gestation length (days)” instead of “Farrowing rate (%)”. We have corrected this error in the revised version. Additionally, the data duplication in the fifth and sixth rows has been reviewed and corrected to reflect the accurate values. We sincerely apologize for these oversights and appreciate the reviewer’s careful attention.

We sincerely thank Reviewer 4 for their thorough and constructive comments, which have significantly improved the clarity, accuracy, and scientific quality of our manuscript. We also sincerely apologize for the errors in the original submission and are grateful for the opportunity to revise and correct them. Your valuable suggestions have greatly enhanced the overall quality and impact of our work.

Round 2

Reviewer 1 Report

Comments and Suggestions for Authors

Dear Authors, thanks for sufficient improvement and addressing to the comments. Authors are further suggested to please note that abstract and each table/figure are dealt as separate entity. Table/figure can explain the things on their own. Previously suggested HPS abbreviation, for example, in table title can be written as full form. If HPS (or any other abbreviation) is within the table body, explain it in the footnote. Please go through every table.

Regards

Author Response

We would like to express our sincere appreciation to Reviewer 1 for the valuable suggestions during this revision. We have carefully considered and addressed all remaining comments, which have further improved the quality of the manuscript. Detailed point-by-point responses are provided below.

Comments: Dear Authors, thanks for sufficient improvement and addressing to the comments. Authors are further suggested to please note that abstract and each table/figure are dealt as separate entity. Table/figure can explain the things on their own. Previously suggested HPS abbreviation, for example, in table title can be written as full form. If HPS (or any other abbreviation) is within the table body, explain it in the footnote. Please go through every table.

Response: Thank you for your valuable suggestion. We have updated all table titles to include the full term “hyperprolific sows (HPS)” instead of using the abbreviation alone. Specifically, these revisions were made at line 274 for Table 2, line 287 for Table 3, and line 302 for Table 4. The abbreviation “HPS” is subsequently used in the body of each table and is clearly defined within the respective table titles. Additionally, the abbreviation used in the body of Table 1 has been explained in the corresponding footnote (line 134). We have carefully checked each table to ensure compliance with this formatting recommendation.

We sincerely thank Reviewer 1 once again for the valuable suggestions and hope that the revised manuscript meets the journal’s standards for publication.

Reviewer 3 Report

Comments and Suggestions for Authors

The authors have significantly improved the manuscript, taking into account all comments, and therefore I accept the article in its current form.

Author Response

We sincerely appreciate Reviewer 3’s positive assessment and recommendation for acceptance. Thank you very much for your valuable time and support throughout the review process.

Reviewer 4 Report

Comments and Suggestions for Authors

The authors addressed my concerns.

Author Response

We sincerely appreciate Reviewer 4’s acknowledgment that our revisions have addressed all concerns. Thank you for your time and valuable input.